# The Influence of Mortar’s Poisson Ratio and Viscous Properties on Effective Stiffness and Anisotropy of Asphalt Mixture

**DOI:** 10.3390/ma15248946

**Published:** 2022-12-14

**Authors:** Marcin D. Gajewski, Jan B. Król

**Affiliations:** Faculty of Civil Engineering, Warsaw University of Technology, 00-637 Warsaw, Poland

**Keywords:** asphalt mixture, effective stiffness modulus, anisotropy, internal structure, image analysis

## Abstract

This paper presents the results of a research study and analysis conducted to determine the degree of anisotropy of asphalt concrete in terms of its initial elastic properties. The analysis of asphalt concrete was focused on determining the effective constrained stiffness modulus in three mutually perpendicular directions based on the finite element method. The internal structure of the asphalt concrete was divided into the mortar phase and the mineral aggregate phase. Static creep tests using the Bending Beam Rheometer were conducted for the mortar phase to fit the rheological model. The aggregate arrangement and orientation were analysed using an image analytical technique for the mineral phase. The Finite Element Method (FEM) meshes were prepared based on grey images with an assumption of plane strain in 2D formulation. Using the FEM model, the tension/compression tests using selected characteristic directions were conducted, and the effective constrained stiffness moduli were estimated. This study showed a dominant horizontal direction for all coarse aggregates resulting from the normal force of the road roller and paving machines during laying and compaction on a road site. Depending on the values of the mortar’s mechanical parameters and the load direction, the effective stiffness modulus might differ by ±20%. Based on the FEM analysis, this result was proven and commented on through an effective directional modulus evaluation and a presentation of internal stress distribution. Depending on the shape and orientation of the aggregates, it was possible to observe local “stress bridging” (transferring stresses from aggregate to aggregate when contacting). Moreover, the rheological properties of the mortar were considered by assuming two limiting situations (instantaneous and relaxed moduli), determining the bands of all possible solutions. In the performed FEM analysis, the influence of the Poisson ratio was also considered. The analysed asphalt concrete tends to be isotropic when the Poisson’s mortar ratio is close to the value of 0.5, which agrees with the physical expectations. The obtained results are limited to particular asphalt concrete and should not be extrapolated to other asphalt mixture types without prior analysis.

## 1. Introduction

The load-carrying capacity of a flexible pavement depends on the parameters of the subbase layer and the thickness, stiffness, and the Poisson ratio of asphalt layers. Various asphalt mixtures are used for pavement layers, and mechanical parameters depend on their internal structure. In stone mastic asphalt (SMA) and porous asphalt (PA) mixtures, the stresses in the material arise mainly in the stone-to-stone contact zone [1]. However, in asphalt concrete (AC), stress also appears in the mastic/mortar phase and is transferred to the mineral skeleton [2]. The behaviour of asphalt concrete is closely related to its internal structure. Understanding the relationship between the internal structure and the mechanical properties of asphalt concrete is very important for modelling road pavement properties [3]. For this purpose, research works have been conducted, based mainly on laboratory test methods, and attempts are made to model the viscoelastic properties of the asphalt mixture using the finite element method. Currently, numerical methods are mainly focused on heterogeneous multi-level modelling, which allows the transfer of bitumen properties to the properties of the final material, i.e., asphalt mixture [4,5,6]. The proper separation of the composite structure components is very important and affects the proper selection of meshes for FEM modelling. In terms of the asphalt mixture structure, three main phases forming the composite material can be mentioned, i.e., mineral aggregate, bitumen, and air voids. On the other hand, the aggregate phase consists of coarse aggregate, fine aggregate, and mineral filler. However, such a division of structures is impossible for FEM modelling because it is very difficult to separate, e.g., bitumen from filler and fine mineral aggregate. That is why asphalt mixture, mastic, mortar, and bitumen are most often mentioned in empirical and experimental works [7,8]. The conclusions from the research [9,10] indicate that the LVE properties of bituminous composites can be predicted based on the model within the acceptable error range. However, to predict the viscoelastic properties of the mixture, it is necessary to construct the best combination of matrix and infill. The results of different types of structures, including mortar, show that the total amount of mortar significantly impacts the final rheological behaviour of the asphalt mixture [10].

The rheological properties of asphalt mixture are complex and the constitutive relationships between stress and strain tensors can be linear or nonlinear depending on the strain amplitude. Some studies [11,12] have shown that both bitumen and asphalt mixtures have linear behaviour at low strain levels, while other studies [13,14,15] have shown that asphaltic materials exhibit nonlinear behaviour at high strain levels. Researchers generally treat this material as isotropic when considering the properties of asphalt mixtures. However, studies have shown that, in terms of the internal structure, the material can exhibit some inhomogeneity which can lead to anisotropic properties, especially in terms of aggregate arrangement and distribution caused by paving and rolling [16,17,18]. Moreover, laboratory compaction methods of asphalt mixtures affect the distribution of grains in the test sample. For example, in the initial phase of impact or gyratory compaction, there are high pressures on coarse aggregate grains, which decrease with increasing compaction cycles, and the contacts between aggregate particles tend to homogenise in the compaction process. Flat particles are subjected to greater stress than shaped particles, and eventually, these elongated grains break during compaction [19]. In addition, the anisotropy of the material may relate to the elastic and plastic properties and the type of elastic anisotropy is not always the same as the type of plastic anisotropy. In general, it can be concluded that the elastic anisotropy determines the material’s behaviour in the first phase of deformation, and the plastic anisotropy may determine the nature of the failure modes. We can introduce the term “initial elastic anisotropy” because the type of anisotropy during the deformation may change (especially when deformation is significant). In addition, a big impact on the results of effective stiffness moduli is the value of the mastic Poisson ratio [20]. A small change in the Poisson’s ratio can affect up to 20% change in the complex modulus value [1]. However, applying a particular level of the Poisson’s ratio is quite problematic in the absence of specific experimental studies [21]. The influence of anisotropic properties can also affect fatigue life, as observed in [22] in the case of an unbounded pavement structure layer.

## 2. Scopes and Objective

This study aims to determine the degree of anisotropy of asphalt concrete in the initial elasticity range. In the scope of this study, an image analysis and a finite element method (FEM) were used for asphalt concrete modelling. The image analytical technique was used to study the mineral aggregate grains’ arrangement. Due to the bitumen content, mortar is a material with rheological characteristics that has been confirmed in flexural creep tests using a Bending Beam Rheometer (BBR). Therefore, a mortar three-parameter standard model was adopted to analyse the linear viscoelastic properties.

The originality of this work lies in the full coupling of image analytical methods with FEM numerical modelling. This means that, based on the image analysis, not only the characteristic features of the mineral or mineral-asphalt mix can be determined, such as grain orientation and the shares of grains of various types or their relative shape, but the very same image is also prepared to generate the FEM mesh. For this purpose, a program written in Wolfram Mathematica was created. Based on the FEM analyses, it is possible not only to determine the directional properties in the asphalt mixture but also to observe local stress states and the way of transferring different loads in the case of contacting and non-contacting aggregate grains (so-called “bridging effect” in the case of contacting aggregates). Another original element is the approach to considering the material’s viscous properties. Instead of modelling the material properties at a selected frequency (load rate), a different approach was chosen. Based on the constitutive model of viscoelasticity (standard model), the limit moduli of elasticity (instantaneous and relaxed) were determined, and, for them, calculations of effective stiffness modules were performed to specify the limit values in the entire frequency range (loading velocity).

## 3. Materials and Testing Methods

### 3.1. Specimen Preparation

Asphalt concrete with a maximum aggregate size of 16 mm (AC 16 binder layer PmB 25/55-60) was used for testing and simulations. The asphalt mixture was produced in an asphalt plant and placed on a road section in one layer with a thickness of 100 mm. The gradation curve and the chemical and basic properties of the materials used in the asphalt concrete are presented in Figure 1 and Table 1 and Table 2. From the same road surface in different locations, five cores for testing were drilled from the compacted pavement layers and trimmed. Each core was divided into two subsamples. The cores in the horizontal and vertical planes were cut to scan the surface of the asphalt mixture structure. The structure images were acquired with a computer scanner. The lateral surface, the top, and the cylindrical samples’ basis were registered, and those cross-sectional structural images were used for image analysis (40 cross-sectional planes in total) and FEM meshing. The image processing methodology and constitutive modelling are described in the following sections.

At the same time, for mixture production, virgin constituent materials, such as bitumen and minerals, were collected for the preparation of a mortar for stiffness measurements. With asphalt binders, filler (89% passes #0.063, 100% passes #0.125), and fine aggregates of particle size up to 4 mm, an asphalt mortar was prepared for the BBR creep tests to fit the rheological model.

### 3.2. Image Analysis

In the testing and modelling of asphalt mixtures’ properties for an internal structure assessment, the methods used are often destructive (e.g., flat cross-section method) and non-destructive methods (e.g., X-ray computer tomography) [23]. Here, the flat cross-section method was applied. The cores were collected from a road section and fine trimmed in a laboratory. The surface of the trimmed core was used for image acquisition and structure analysis. The cores were drilled from the binder course at right angles to the course layer but without information about the ride direction of the road roller machine. Based on previous research findings presented in [18], it was supposed that the orientation of the aggregate particles was equal in sections *zx* and *zy* (Figure 2) regardless of the direction of the road roller machine moving. The lateral surface of the core was used to prove the assumption about the lack of impact of the roller direction on the aggregate particles’ orientation. Figure 2 shows the orientation of the sample according to the axes.

The image of the lateral sample surface was acquired with a resolution of 300 dpi using a linear computer scanner mounted on an image registration device. The device for image acquisition from a lateral surface of the core is described in [18]. The plane sections of the top and bottom specimens were scanned with a flat computer scanner. The acquired images were analysed using an image analysis computer program, ImageJ (v.1.52a) (National Institutes of Health, Bethesda, MD, USA). After image binarisation, separate grains with a size larger than 2 mm were extracted, and the ellipse was adjusted on them in order to determine the minimum and maximum diameters. Based on the fitted ellipse, a grain orientation was calculated as an angle inclination of the maximum diameter to the horizontal axis. The investigated area of the lateral sample surface was divided into eight equal regions and marked on four different colours, as shown in Figure 3. The four colours designate four distinct regions in the period of 180°. Each colour marked region of the lateral specimen surface has been numbered, e.g., 1 and 1’, where 1’ corresponds to direction 1 in the period of 180°. The arrangement of the analysed areas on the lateral surface of the sample is shown in Figure 2, and in the development of the lateral surface in Figure 3. Cut grains placed onto the bounded areas are marked with a grey colour. These grains were not considered in the analysis to avoid boundary effect regarding size effect. The dominant orientation in each coloured grain group was analysed.

The calculated values of the angle inclination of separate grains are ranked in frequency and are shown in Figure 4. The dominant mode enables the concluding arrangement of aggregate grains in the asphalt mixture.

Based on the analyses of the mean value (Figure 4e) of frequency distribution of different angle-orientated grains, it can be stated that there is a dominant mode for angle 0° (in the period of 180°) for all grains independently of the regions. The dominant horizontal orientation (0°) of grains is probably because of the effect of normal force during compaction by a roller machine on a road site. A high amount of oriented grains can be observed in the first direction (1/1′—blue, Figure 4a), and a low amount of grains can be observed in the fourth direction (4/4′—yellow, Figure 4d). These directions are located next to each other (Figure 3), which excludes the domination of one of them because the road roller machine moves parallel to the road axis, and not across. It should be stated that, based on the orientation of the grains, it is not possible to find the road roller direction during the compaction process, and this allows the selection of one random cross section as a representation for both sections *xz* and *yz*.

To prepare the FEM mesh, the vertical cross section *xz*, which is equivalent to the *yz*, and the horizontal cross section *xy* of the cores were used. Two computer bitmap files were generated in 256 grey-value mode based on two selected images and were reduced from the original size of 830 × 830 pixels to 415 × 415 pixels. To distinguish mineral materials (melaphyr and dolomite), grey plot profile was analysed. Figure 5 shows one of the analysed images where, for melaphyr, the grey value is from 120 to 180, and for dolomite, grey value is over 200. For the mortar, the grey value is from 0 to 70.

### 3.3. Constitutive Modelling

Mortar is a material with rheological properties. For its material description, the standard model (serial connection of the Kelvin–Voigt model with elastic element) is applied with the mechanical interpretation, as shown in Figure 6.

It should be underlined that this is one of the simplest models. However, for assessing the limit of Young’s modulus, it appears to be adequate. The parameters for this model are based on the experimental tests performed using a BBR at 10 °C. The test result of the rapid beam loading, which is then kept constant for a certain time, can be considered a result of creep tests. Therefore, it is necessary to determine the creep function, which in the case of the standard model, has the following form:(1)$$\tilde{J}\left(t\right)=\frac{E+E_{1}}{EE_{1}}\left(1-\frac{E}{E+E_{1}}e^{-\frac{tE_{1}}{\eta_{1}}}\right)H\left(t\right).$$
where:

$\tilde{J}\left(t\right)$—creep function;E, $E_{1}$—parameters of the model as shown in Figure 6 with stiffness dimension;$\eta_{1}$—parameter of the model as shown in Figure 6 with viscosity dimension;t—time;$H\left(t\right)$—Heaviside step function.

For the applied rheological model of solid, the characteristic limits of moduli can be determined. In the case of rapidly changing loads, the modulus aims for the value $E_{0}=E.$ On the other hand, in the case of static loading, the limit of the modulus (called relaxed modulus) is equal to $E_{r}=\frac{EE_{1}}{E+E_{1}}$. Based on the creep Equation (1) and the solution of a three-point bended beam loaded with a concentrated force in the middle of the span, it is possible to derive the beam’s deflection function in the following Equation (2):(2)$$w\left(t\right)=\frac{\tilde{J}\left(t\right)Pl^{3}}{4bh^{3}}.$$
where:

$\tilde{J}\left(t\right)$—creep function;P—force loading of the BBR beam sample;l—span of the BBR beam sample;b—width of the BBR beam sample;h—height of the BBR beam sample.

The above formula is the starting point for determining the parameters E, $E_{1}$, and $\eta_{1}$. Determination of these parameters based on the BBR test results is possible with the application of nonlinear optimisation techniques. In the considered case, the Mathematica software with the standard procedure Nonlinear Model Fit was applied. The mortar’s material parameters were determined in three cases:All three parameters were obtained from nonlinear fitting with an application of the same weights for all data from the BBR test.All three parameters were obtained from the nonlinear fitting procedure with the assumption that the weights of the data for t→0 are significantly higher than for the other data (it was assumed that the weight of the first measured point is 50 times higher than the weight of the point for t=240 [s]).The only parameter $\eta_{1}$ was estimated based on the nonlinear fitting procedure, and two other parameters were established based on the data for the minimum and maximum relaxation time recorded in the test performed on the mortar beam.

The obtained standard model parameters are shown in Table 3. The parameters of the standard model in cases (ii) and (iii) are very similar. They differ significantly from only the parameters in case (i).

The compatibility of the adopted model (with the parameters like those shown in Table 3) with the BBR test results in a creep test is shown in Figure 7.

As mentioned above, based on the standard model, the two characteristic mortar material moduli, which correspond to the rapidly changing and static loads, can be specified. Therefore, to determine the effective properties of the aggregate-mortar composite, a constitutive model of isotropic linearly elastic materials is applied with the moduli in the case of the mortar being equal to $E_{r}$ and $E_{0}$, respectively. Performing the calculations for these limit cases ensures that all the other behaviour of the material is located between them.

The constitutive relationship of isotropic linearly elastic materials is as follows:(3)$$\sigma =2Ge+K\left(\mathrm{tr}\varepsilon \right)I,$$
where
(4)$$G=\frac{E_{\alpha }}{2(1+\nu_{o})},\;K=\frac{E_{\alpha }}{3(1-2\nu_{o})},$$
and α=0,r. In Equation (3), **σ** stands for the stress tensor, e is a deviatoric part of the strain tensor (e.g., $e=\varepsilon -\left(\mathrm{tr}\varepsilon \right)I/3$), “tr” operator is understood as the trace of the tensor, while I is an identity second-order tensor. For example, in the case of the analysed mortar parameters determined for case (i), in a constitutive relationship (3), the value $E_{0}$ = 505 [MPa] or $E_{r}$ = 31 [MPa] and the Poisson ratio $\nu_{o}$ are substituted.

In the constitutive modelling of aggregate materials, the isotropic linearly elastic Hooke’s material model is assumed with a relationship like in Equation (3). The assumed material parameters are presented in Table 4.

## 4. Finite Element Model Analysis

The grey-scale graphic files in tiff format with the dimensions of 415 × 415 pixels were prepared based on the selected images. It is assumed that the black colour (gs = 0) refers to mortar, the grey colour (gs = 98) refers to dolomite, and the white colour (gs = 255) refers to melaphyre (cf. Figure 5 and Figure 8).

Based on these two images (Figure 8), two FEM meshes were prepared with the assumption that the plane strain state is valid. The FEM meshes are regular and consist of 415 × 415 plane elements with linear shape functions. The ABAQUS/Standard (v.6.14) FEM software (ABAQUS Inc., Palo, Alto, CA, USA) was used to solve the boundary-value problems.

A significant difficulty was generating the input file for the ABAQUS (v.6.14) program taking into account the properties of the three component materials based on the obtained images. In order to realise this goal, a dedicated program in Wolfram Mathematica (v.9.0) (Wolfram Research, Inc., Champaign, IL, USA) was written to solve this problem. For each pixel on the image, a value of grey scale is assigned for the particular element and some specific material properties; then, on that basis, a set of elements is created which allows the generation of an input file for the ABAQUS. Figure 9 presents the created FEM meshes for all asphalt mixture phases in particular planes.

### 4.1. Boundary Conditions

To establish the effective constrained moduli of material, the compression of the elements with mesh (Figure 9) in the direction of the axes *x*, *y*, and *z* with appropriate boundary conditions are solved as 2D plane strain tasks. For example, compressing the FEM mesh created from an image in *xy*-plane in the x-direction on the boundary AD (Figure 10), it is assumed that $u_{x}$ = 0 on boundaries AB and CD and that $u_{y}$ = 0, and while on the boundary BC, the non-zero displacement $u_{x}$ is assumed to realise an average compression strain equal to 1.0%. The reaction forces will appear on all boundaries because of the assumed non-zero displacement boundary conditions. For example, on the boundary BC, the reactions can be denoted as $R_{x}^{p}$ (see Figure 10a). On the other hand, in the case of the compression test in the direction *y*, on the boundary AB, it is assumed that $u_{y}$ = 0, and while on boundaries BC and AD, $u_{x}$= 0. Finally, on the boundary CD, the non-zero displacement $u_{y}$ is assumed to realise the compression of the sample with an average strain equal to 1.0%. As a reaction to that displacement in the nodes on the boundary CD, the reaction forces appear, which are denoted as $R_{y}^{p}$ (see Figure 10b).

Similarly, the boundary conditions are assumed in the case of the FE mesh obtained based on the image in the *xz* plane, as shown in Figure 9 (corresponding axes, in this case, are presented in Figure 10 in parentheses with lighter colour). In addition to the notation for the *x*-axis, the “star” is added ($x^{*}$) to indicate the same axis. However, as presented in another image, the calculation of effective constrained modulus in this direction can be regarded as an indicator of the sample result’s representativeness. The error of this estimation allows for the assessment of other results in terms of their adequacy (validation).

### 4.2. The Idea of Estimation of Effective Moduli Values

For tests with the boundary conditions as in Figure 10, the constrained effective moduli were determined based on the following formula:(5)$$E_{i}=\frac{l\sum_{p=1}^{P}R_{i}^{p}}{A\;u_{i}},$$
where i=x,y,z, $R_{i}^{p}$ is a reaction force for the displacement in the i direction in the p node. A is a transverse cross-section area A=hl, and it was assumed that h = 1 [mm]. Respectively on the boundary BC or CD, there are P nodes (in analysed case P = 416).

## 5. Results and Discussion

### 5.1. Effective Stiffness Moduli in an Exemplary Case (ν=0.4)

Taking into account the boundary conditions presented above for each of the FEM mesh shown in Figure 9, two compression tests were performed for each image, thus determining $E_{x}$, $E_{y}$, $E_{x}^{*}$, and $E_{z}$. In all tests, the material parameters for aggregates were assumed as presented in Table 4, while the Young moduli for the mortar were assumed to be like those in Table 3 with the Poisson coefficient equal to ν=0.4. Each of the four boundary value problems was solved with six different sets of data for the mortar (i.e., in the case of mortar material, the data set determined in three optimisation variants was used, assuming an initial stiffness modulus $E_{0}$ or a relaxed one $E_{r}$). The results for the constrained effective stiffness moduli as determined using the Formula (5) are presented as a bar graph in Figure 11, and in Table 5.

Based on the graphs presented in Figure 11 and the results presented in Table 5 and Table 6, a significant difference in the modulus for the *y*-axis direction of the sample can be observed. The module in this direction is more than 17% lower than that in the *x*-axis direction, in the case of the assumption of the relaxed modulus value $E_{r}$ for mortar (for an error level estimated based on the comparison of the results for the moduli in the *x*-direction and $x^{*}$ at 6.5%). In this case, the modulus $E_{z}$ is comparable to the value of the modulus $E_{x}$ at the evaluated error level. A similar result is also obtained in the case of the adoption of the initial modulus $E_{0}$ value for mortar. However, this time, the modulus $E_{y}$ is smaller by about 9% (on an error estimation of 4%). These results indicate that the degree of anisotropy of the material in the presented case decreases when the mortar stiffness increases, given the observation that, if the mortar stiffness approaches the stiffness of the aggregates, the composite material properties will approach the isotropy. These results correspond with the conclusions from other works where the shape of the grains and their arrangement in the mixture are analysed [19,25,26]. This becomes particularly important in mixtures where there are flat grains that determine the initial anisotropy, since the position of large-size particles is difficult to be changed in the compaction process [27]. The compaction scale effect may also influence the aggregate arrangement, for example, a laboratory scale mould may potentially influence the curve of impact compaction and grain orientation [18,28].

In Figure 12 and Figure 13, the contour graphs of the principal strains and the equivalent Mises stresses are presented for an exemplary case of a mortar described with the parameters $E_{r}$ = 31.47 [MPa] and ν=0.4 for the images in the *xy* (*xz*) plane compressed, respectively, in the x and y (x and z) directions. The extreme values of the maximum and minimum principal strains are presented in Table 7. It is worth emphasising that, for the average compressive strain of 1%, the local maximum and minimum principal strain exceeds 30% and −20%, respectively. Additionally, in Table 8, the maximum Mises equivalent stresses in the mortar and aggregate are presented. In the mortar, the maximum Mises stress exceeds 10 MPa, while the aggregate’s value is close to 100 MPa. In one case, the maximum Mises stresses are greater than 187 MPa (in this case, there is a clear contact between the components of the aggregate). The contour graphs shown in Figure 12 and Figure 13 and in Table 7 and Table 8 present the results for one chosen variant of the mortar material parameters, but the results obtained in other cases are very similar, and they are not presented here in detail.

The contour graphs of the equivalent Mises stresses in Figure 13 show local stress concentrations in the spaces between the grains of the aggregate. These graphs can be treated as an illustration of the load transfer between different elements of the composite; this has also been confirmed in [3] which predicts that stress concentration at the aggregate–asphalt interface tends to induce a higher percentage of damaged elements, especially adhesive-damaged elements. These plots should be analysed together with the images of the sample in the *xy* and *xz* planes, respectively.

### 5.2. The Influence of the Poisson Ratio

For these calculations, the mortar’s material data are selected as in the second row of Table 3. Figure 14 contains the effective stiffness moduli $E_{x}$, $E_{y}$, $E_{x}^{*}$, and $E_{z}$ as the functions of the Poisson ratio for the mortar’s stiffness modulus $E_{r}$ (Figure 14a) and $E_{0}$, respectively (Figure 14b). These graphs show that, when ν→0.5, then the difference between the moduli is reducing.

Additionally, in the form of bar charts from Figure 15, Figure 16 and Figure 17**,** the percentage differences between the effective moduli in different directions in relation to the modulus $E_{x}$ are also provided as a function of the Poisson ratio.

As presented in Figure 17, the bar chart $100\left(E_{x}^{*}-E_{x}\right)/E_{x}$ can be treated as previously noted as an error level estimation. The error estimation for the composite stiffness modulus in the case of the mortar characterised with $E_{r}$ (Figure 17a) has the maximum value of ν=0.3 and is larger than 7%. With an increase in the Poisson ratio ν→0.5, this error is reduced to a value smaller than 4%. In the mortar case with the modulus $E_{0}$, the error estimation decreases from a 5% value to zero, and when ν→0.5, it changes its sign. Taking into account the values of these error estimations, it appears indisputable that stiffness in the direction of the *y*-axis is different, both in the case of the mortar characterised by $E_{r}$ and by $E_{0}$ (cf. Figure 15a,b). In the case of other directions, the differences in the moduli values are at the level of error estimation.

## 6. Conclusions and Final Remarks

When formulating the conclusions based on the image and FEM analyses, it should be noted that they cannot be generalised to other types of asphalt mixtures and aggregates because their specific structure can significantly affect the results obtained in this work.

The analysis of aggregate distribution using image analysis allows the assessment of the dominant mode in the direction of the coarse mineral grains in the asphalt mixture. This study has shown that it is possible to analyse only coarse aggregate grains while the fine aggregate (sand) and filler must be considered in the mastic or mortar phase. For FEM modelling, dividing the asphalt composite phases as coarse aggregate and mastic or mortar is possible and sufficient.Based on the analyses of the frequency distribution of grain orientation, there is a dominant horizontal direction for all coarse aggregate. The dominant horizontal orientation (0°) of the grains results from the normal force of the roller during compaction on road sites. Additionally, this study has shown a lack of impact of the roller moving direction on the aggregate particle’s orientation.Based on the determined asphalt concrete effective stiffness moduli (using FEM), a significantly lower modulus in the direction of the *y*-axis is observed. Depending on the values of the mortar’s mechanical parameters, the modulus value $E_{y}$ is smaller, even when compared to 20% of the $E_{x}$ modulus value. With a rather complex process of computational model preparation and limitations obvious from a statistical point of view, the authors recommend extreme caution when formulating conclusions about the initial anisotropy of the material. In the case of other directions, the differences in the moduli values are at the estimated error level.From the FEM analysis, a significant effect of the Poisson’s ratio on the analysed asphalt mixture stiffness and its degree of anisotropy is shown. At ν→0.5 (i.e., when the mortar material tends towards incompressibility), the composite material tends towards isotropy.Interestingly, the analysis shows the influence of viscous material properties, i.e., the degree of anisotropy depends on the deformation rate (loading speed).

From a practical point of view, the present results can be used to understand better the working mechanisms of a composite composed of asphalt mortar filling and much stiffer aggregate grains. They make it possible to notice that the internal arrangement of grains determines not only the working conditions and possible failure mechanisms but also the degree of initial anisotropy. From an engineering point of view, the way that the mix is placed/compacted produces directional material properties has always been intuitively understood, and the detailed analysis presented in this paper confirms these observations.

Moreover, different ways of transferring loads can be noticed based on the analysis of local stress states inside the heterogeneous structure of the mineral-asphalt mixture. Analysis proves that local strains within a heterogeneous structure can be more than thirty times higher than global average strains. As expected, these values significantly depend on the aggregate’s morphological properties and the grains’ arrangement inside the structure. The translation of these strain fields into the development of local damage at the micro-scale (material fatigue) and macro-scale (failure mechanisms) levels is immediate. According to the authors, this is one of the elements that may contribute to understanding the development of the fatigue process of mineral-asphalt mixtures. Performing the analysis as proposed in this article with the use of image analytical methods and the finite element method seems to be a process that can be automated and be faster than determining the full fatigue characteristics of the material. Of course, it is impossible to replace experimental fatigue studies, but understanding the internal mechanisms of the material’s behaviour may significantly reduce them.

In summary, attention should be paid to the limitations that might have affected the obtained results. The biggest limitation is that the analysis was performed in 2D instead of 3D. Conducting a two-dimensional analysis is a simplification both in the analytical and experimental setup levels. In the analytical level, the resignation from one dimension significantly simplifies the formulation, and in FEM analysis, it is possible to map the internal structure of the mixture in detail while maintaining a sufficient number of finite elements (415 × 415 = 172,215 elements). At the same level of detail, a 3D analysis would require a mesh of over 71 million elements.

The above comments regarding the limitations of the presented formulation are, at the same time, an indication of directions for further research. Of course, the correctness of this formulation should be verified by comparing the influence of 2D and 3D modelling on the obtained results (treating 3D models as a reference point). In addition, for 2D modelling, a procedure for preparing the surface for scanning should be proposed so that the air void ratio remains at the appropriate expected level. A natural direction for extending the methodology presented in this work is to repeat the analysis with other characteristic asphalt mixtures found in road construction, such as SMA and porous asphalt, while considering different asphalt binders (including highly modified binders).

## Figures and Tables

**Figure 1 materials-15-08946-f001:**
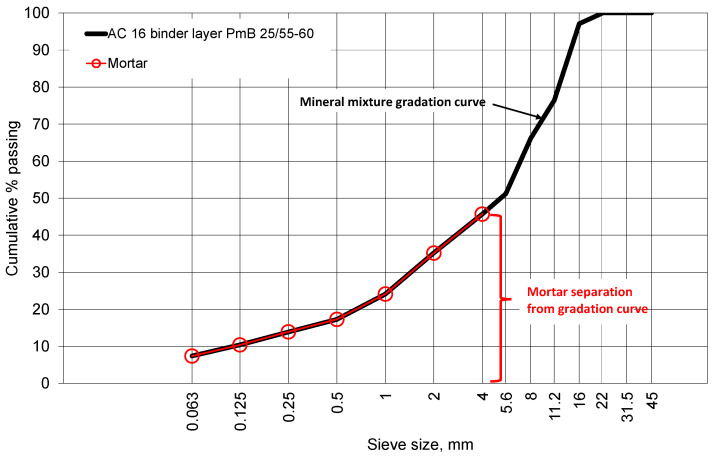
Asphalt concrete and mortar asphalt gradation curve.

**Figure 2 materials-15-08946-f002:**
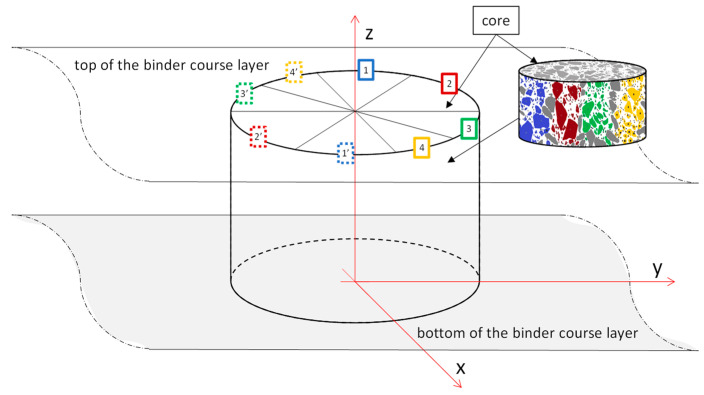
Orientation of the sample according to the axes.

**Figure 3 materials-15-08946-f003:**
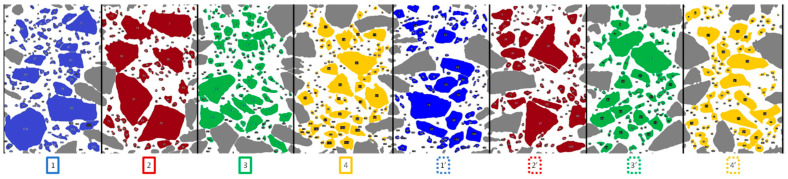
Developed area of the sample lateral surface shared by eight specific areas for the orientation test.

**Figure 4 materials-15-08946-f004:**
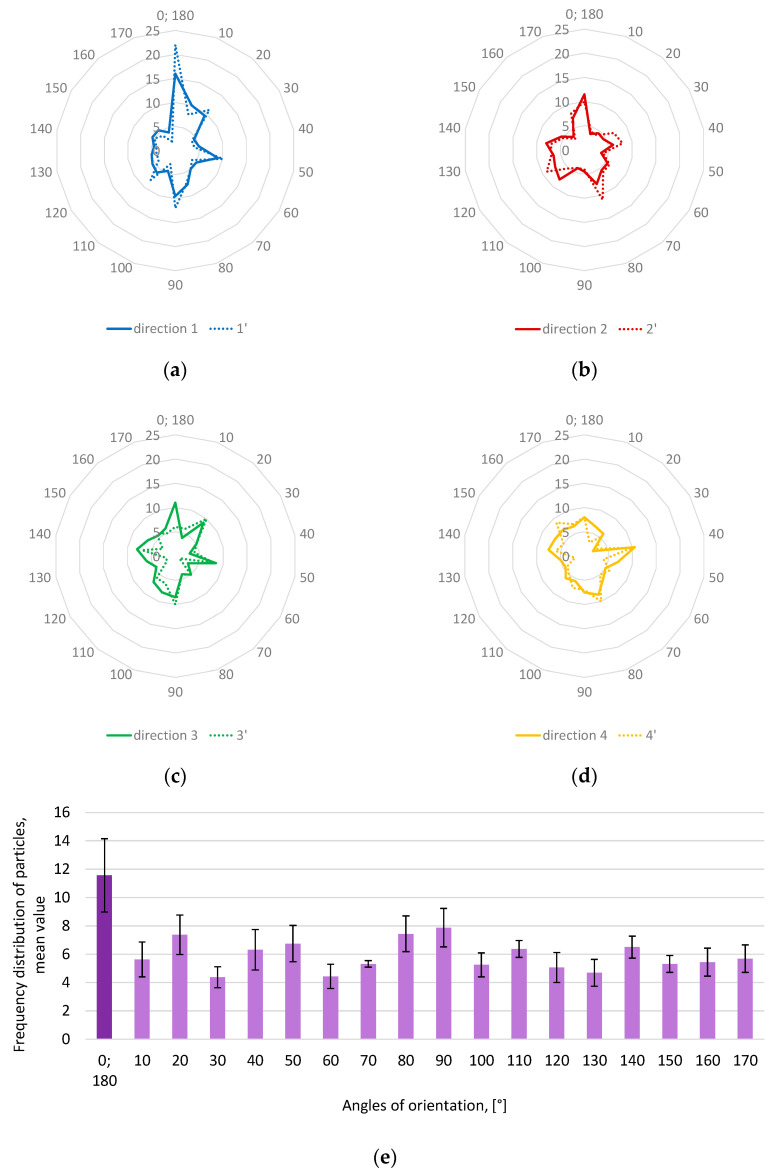
Frequency distribution of orientated grains with different orientations: (**a**) direction 1/1’; (**b**) direction 2/2’; (**c**) direction 3/3’; (**d**) direction 4/4’; and (**e**) mean value.

**Figure 5 materials-15-08946-f005:**
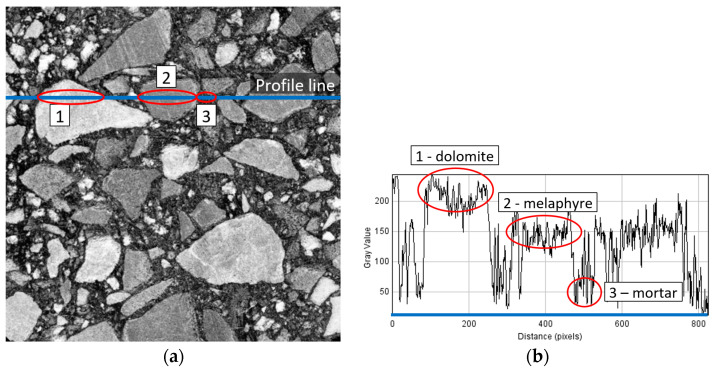
Cross section of asphalt mixture: (**a**) gray image with example of profile line, (**b**) grey plot profile.

**Figure 6 materials-15-08946-f006:**
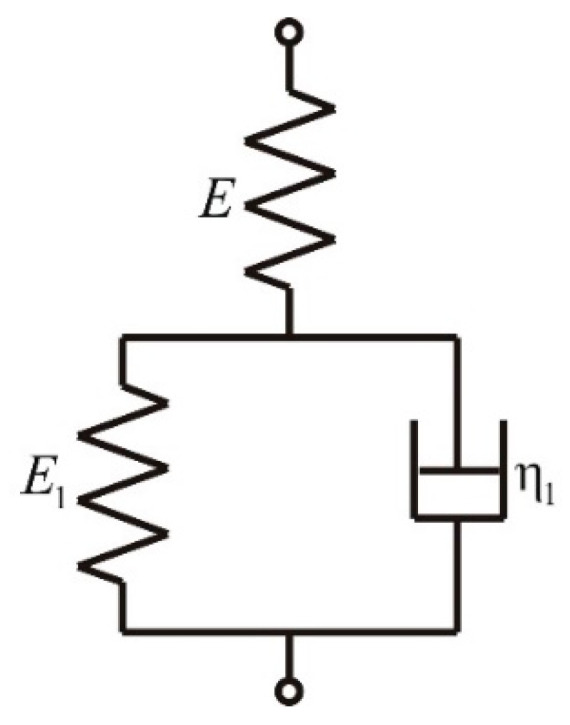
Schematic diagram of the standard model with the component properties in symbolic designation.

**Figure 7 materials-15-08946-f007:**
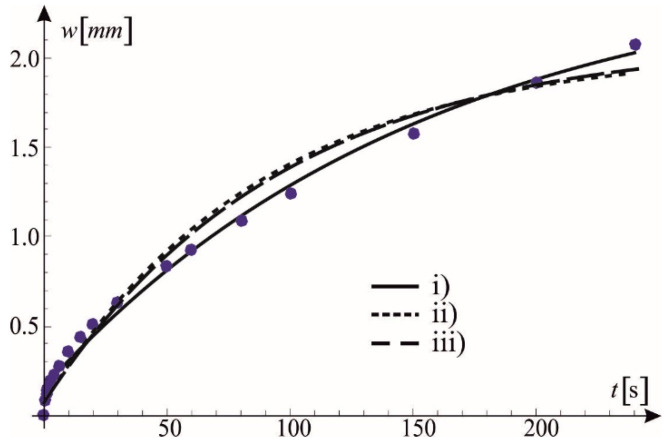
Mortar creep test vs. predictions of the standard model with three sets of parameters.

**Figure 8 materials-15-08946-f008:**
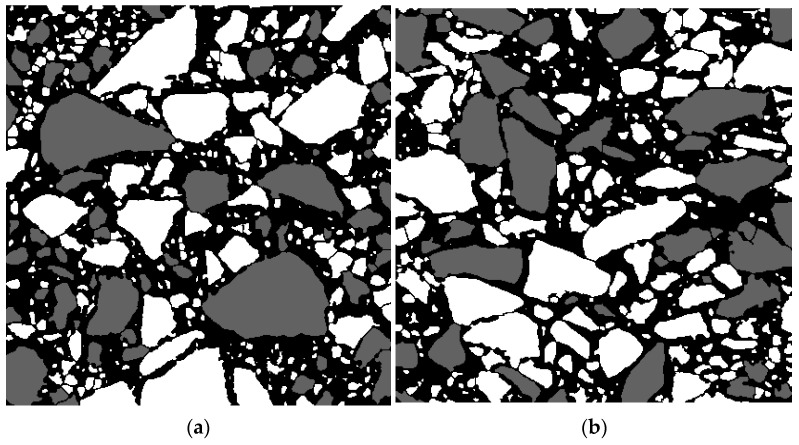
Grey images: (**a**) scan in *xy* plane; (**b**) scan in *xz* plane.

**Figure 9 materials-15-08946-f009:**
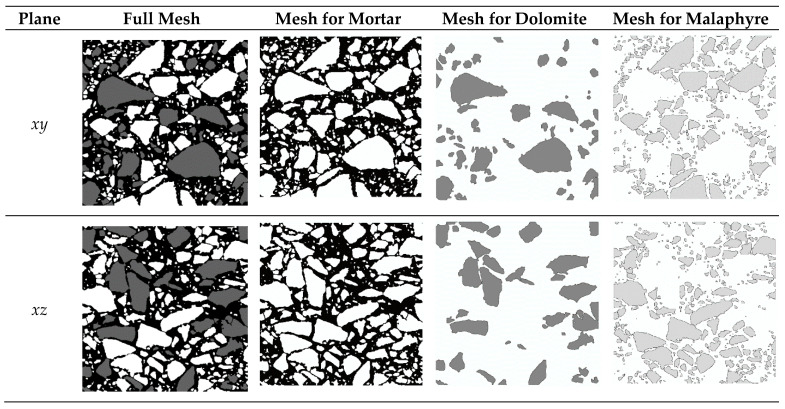
FEM mesh generated based on the source image for all asphalt mixture phases.

**Figure 10 materials-15-08946-f010:**
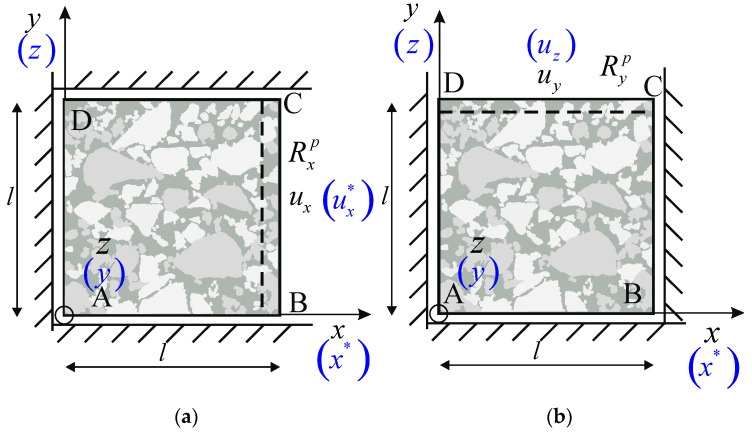
The schematic view of the boundary conditions assumed for the determination of the constrained effective moduli: (**a**) boundary BC, (**b**) boundary CD.

**Figure 11 materials-15-08946-f011:**
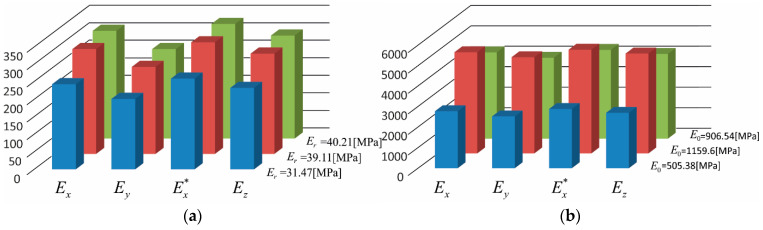
The values of the effective stiffness moduli determined for particular directions for different mortar material parameters (ν=0.4): (**a**) for relaxed moduli, and (**b**) for initial moduli.

**Figure 12 materials-15-08946-f012:**
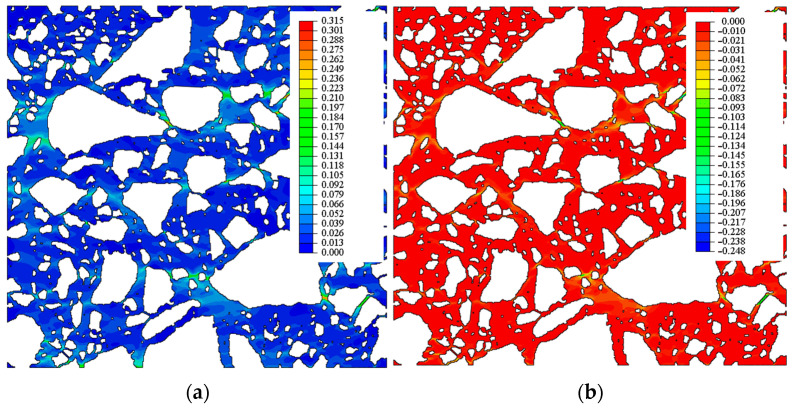
Contour graphs of principal strains: (**a**) maximum and (**b**) minimal in the mortar.

**Figure 13 materials-15-08946-f013:**
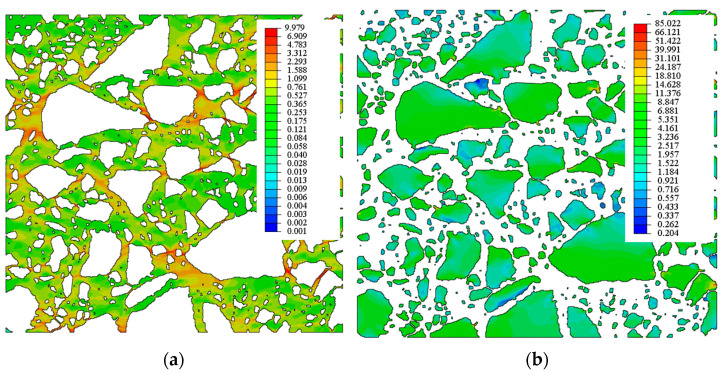
Contour graphs of the equivalent Mises stresses in (**a**) mortar and (**b**) aggregate.

**Figure 14 materials-15-08946-f014:**
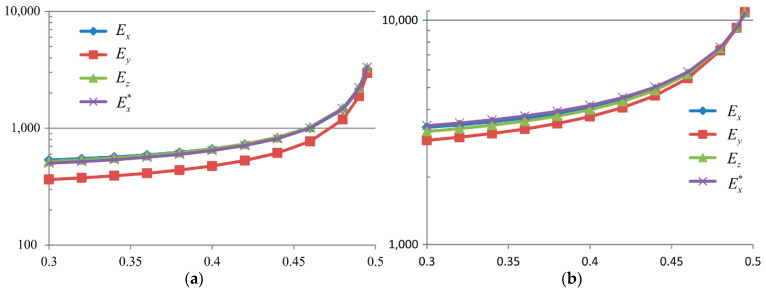
The influence of the mortar’s Poisson ratio on the value of the effective constrained moduli of asphalt concrete with the assumption of the mortar’s Young modulus as (**a**) $E_{r}$ and (**b**) $E_{0}$ /the (i) variant/.

**Figure 15 materials-15-08946-f015:**
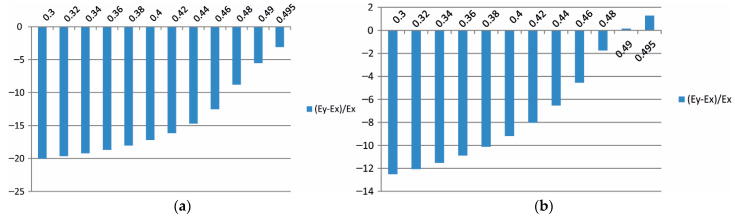
The influence of the mortar’s Poisson ratio on the value of $E_{y}$ modulus of asphalt concrete with assumption of the mortar’s Young modulus as (**a**) $E_{r}$ and (**b**) $E_{0}$/the (i) variant/.

**Figure 16 materials-15-08946-f016:**
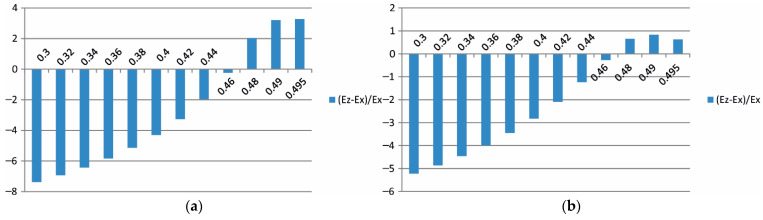
The influence of the mortar’s Poisson ratio on the value of $E_{z}$ modulus of asphalt concrete with the assumption of the mortar’s Young modulus as (**a**) $E_{r}$ and (**b**) $E_{0}$/the (i) variant/.

**Figure 17 materials-15-08946-f017:**
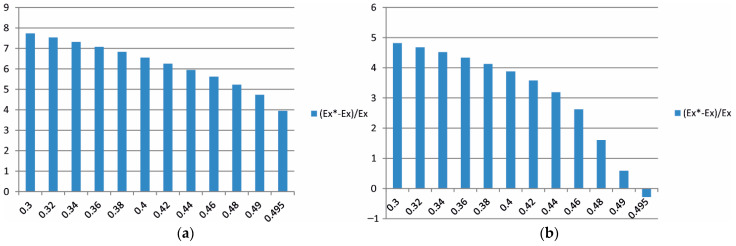
The influence of the mortar’s Poisson ratio on the value of $E_{x}^{*}$ modulus of asphalt concrete with the assumption of the mortar’s Young modulus as (**a**) $E_{r}$ and (**b**) $E_{0}$/the (i) optimisation variant/.

**Table 1 materials-15-08946-t001:** Composition and chemical and petrographic origin of used materials.

Properties	Unit	% in the Mix	Chemical/ /Petrographic Origin	Density at 20 °C Mg/cm^3^
Mixture Composition				
Filler (0–0.063 mm)	%	4.7	Lime	2.640
Fine aggregate (0.063–4.0 mm)	%	41.7	Basalt	2.930
Coarse aggregate (>4 mm)	%	48.4		
4/8 mm 8/16 mm 11/16 mm			Melaphyre Dolomite Melaphyre	2.696 2.789 2.690
Bitumen	%	5.2	PmB 25/55–60 (SBS modified)	1.015
Adhesive	%	0.2	Amine based	0.980

**Table 2 materials-15-08946-t002:** Volumetric properties of the asphalt mixture.

Properties	Unit	Results
Density of mineral aggregate	g/cm^3^	2.759
Voids in mineral aggregate VMA	%	15.7
Voids filled by binder VFB	%	78.3
Maximum density	g/cm^3^	2.539
Bulk density	g/cm^3^	2.452
Air void content	%	3.4

**Table 3 materials-15-08946-t003:** Parameters for the standard model of mortar determined in three different ways.

	$E=E_{0}$ [MPa]	$E_{1}$ [MPa]	$\eta_{1}$ [MPas]	$E_{r}$ [MPa]
(i)	505	34	5219	32
(ii)	1160	41	3600	39
(iii)	906	40	3829	38

**Table 4 materials-15-08946-t004:** Young’s moduli and Poisson’s ratios for aggregates.

Material	Young Modulus [MPa]	Poisson Ratio [–]	Compression Strength [MPa]	Friction Angle [Degrees]
Dolomite	12,000	0.20	60–160	27–31
Melaphyre *	9000	0.23	130–200	33–40

* The data are assumed based on Yazdani et al. [24].

**Table 5 materials-15-08946-t005:** The values of the effective stiffness moduli determined for particular directions for different mortar material parameters (ν=0.4).

Mortar Stiffness [MPa]		$E_{x}$ [MPa]	$E_{y}$ [MPa]	$E_{x}^{*}$ [MPa]	$$E_{z}$$ [MPa]
$$E_{r}$$	31	243	201	259	233
	39	299	248	319	286
	40	307	255	327	294
$E_{0}$	505	2789	2532	2879	2710
	1160	4949	4704	5072	4886
	907	4218	3956	4340	4144

**Table 6 materials-15-08946-t006:** The percentage difference between values of the particular effective stiffness moduli and the $E_{x}$ modulus for different mortar material parameters (ν=0.4).

Mortar Stiffness [MPa]		$100\left(E_{y}-E_{x}\right)/E_{x}$ [%]	$100\left(E_{x}^{*}-E_{x}\right)/E_{x}$ [%]	$100\left(E_{z}-E_{x}\right)/E_{x}$ [%]
$E_{r}$	31	−17.2	6.6	−4.3
	39	−17.0	6.5	−4.4
	40	−17.0	6.5	−4.4
$E_{0}$	505	−902	3.9	−2.8
	907	−6.2	2.9	−1.8
	1160	−5.0	2.5	−1.3

**Table 7 materials-15-08946-t007:** Maximum and minimal principal strains depend on the particular image and the compression direction.

Image	*xy*		*xz*	
Compression Direction	*x*	*y*	*x*	*z*
$\varepsilon_{\max}$ [%] (for a. s.* 1.0%)	31.5	28.7	35.0	39.2
$\varepsilon_{\min}$ [%] (for a. s.* 1.0%)	−24.8	−15.8	−22.7	−20.2

*—average strain

**Table 8 materials-15-08946-t008:** Maximum values of the Mises equivalent stresses depend on the particular image and the compression direction.

Image	*xy*		*xz*	
Compression Direction	*x*	*y*	*x*	*z*
$\sigma_{z}$ [MPa] (for a. s.* 1.0%) in mortar	10.0	8.8	10.4	9.1
$\sigma_{z}$ [MPa] (for a. s.* 1.0%) in aggregate	85.0	116.5	187.2	83.1

*—average strain

## Data Availability

Data sharing is not applicable to this article.

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
