# Peer review of "The Influence of Mortar’s Poisson Ratio and Viscous Properties on Effective Stiffness and Anisotropy of Asphalt Mixture"

_materials, 2022, doi:10.3390/ma15248946_

Round 1

Reviewer 1 Report

In this study, the distribution of coarse aggregate particles and orientation was analyzed by using the digital images of horizontal and vertical sections of in-situ coring specimens; Next, the difference of modulus in different directions and the influence of Poisson's ratio are analyzed by finite element method. These studies are helpful to understand the influence of asphalt mixture microstructure on macro performance. However, as the research of engineering materials, the paper should further introduce and discuss the value or enlightenment of relevant conclusions for asphalt pavement and mixture design and practical engineering.

Author Response

We thank the reviewer for their appreciation. The last remark addressing the practical inclinations of the proposed methodology was addressed in paragraph 6. The following explanation was added in the revised manuscript:

From a practical point of view, the presented results can be used to understand better the working mechanisms of the composite composed of asphalt mortar filling and much stiffer aggregate grains. They make it possible to notice that the internal arrangement of grains determines not only the working conditions and possible failure mechanisms but also the degree of initial anisotropy. From an engineering point of view, the way the mix is placed/compacted produces directional material properties has always been intuitively understood, and the detailed analysis presented in this paper confirms these observations.

Moreover, different ways of transferring loads can be noticed based on the analysis of local stress states inside the heterogeneous structure of the mineral-asphalt mixture. Moreover, the analysis proves that local strains within a heterogeneous structure can be even more than thirty times higher than global average strains. As expected, these values significantly depend on aggregate’s morphological properties and the grains’ arrangement inside the structure. The translation of these strain fields into the development of local damage in the micro-scale (material fatigue) and macro-scale (failure mechanisms) is immediate. According to the authors, this is one of the elements that may contribute to understanding the development of the fatigue process of mineral-asphalt mixtures. Carrying out the analysis proposed in the article with the use of image analysis methods and the finite element method seems to be a process that can be automated and faster than determining the full fatigue characteristics of the material. Of course, it is impossible to replace experimental fatigue studies, but understanding the internal mechanisms of the material’s behaviour may significantly reduce them.

Reviewer 2 Report

The paper topic is interesting. I suggest some comments as shown below

1-The abstract must include key numerical findings

2-the originality must be clearly stated

3-the limitation of the study must be added

4-the chemical composition of the used materials are missing and must be added

5-it would be better to prepare a tabular form of the parameters that affects the behavior of the asphalt material 

6-the recommendation for the future work must be added

Author Response

Thank you very much for the quick review process and provided comments improving our paper. All of the comments have been reflected in the fixed manuscript.

- The paper topic is interesting. I suggest some comments as shown below

We thank the reviewer for their appreciation.

- The abstract must include key numerical findings

Addressing this remark, the following text was added to the Abstract:

Based on FEM analysis, this fact was proven and commented on effective directional moduli evaluation and presentation of internal stress distribution. Depending on the shape and orientation of aggregates, it is possible to observe local "stress bridging" (transferring stresses from aggregate to aggregate when contacting). Moreover, the rheological properties of the mortar's were considered by assuming two limiting situations (instantaneous and relaxed moduli), determining the bands of all possible solutions. In performed FEM analysis, the influence of Poisson ratio was also considered. Analyzed asphalt concrete tends to be isotropic when Poisson's mortar ratio is close to the value of 0.5, which agrees with physical expectations.

- The originality must be clearly stated

Addressing this remark, the following explanations were added in paragraph 2 of the paper:

The originality of this work lies in the full coupling of image analysis methods with FEM numerical modelling. That means that based on the image analysis, not only the characteristic features of the mineral or mineral-asphalt mix were determined, such as grain orientation, shares of grains of various types or their relative shape, but also the very same image was prepared to generate the FEM mesh. For this purpose, a program written in Wolfram Mathematica was created. Based on FEM analyses, it is possible not only to determine the directional properties in the asphalt mixture but also to observe local stress states and the way of transferring different loads in the case of contacting and non-contacting aggregate grains (so-called "bridging effect" in the case of contacting aggregates). Another original element is the approach to considering the material's viscous properties. Instead of modelling the material properties at a selected frequency (load rate), a different approach was chosen. Based on the constitutive model of viscoelasticity (standard model), the limit moduli of elasticity (instantaneous and relaxed) were deter-mined, and for them, calculations of effective stiffness modules were carried out, specify-ing limit values in the entire frequency range (load speed).

- The limitation of the study must be added

Addressing this remark, the following explanations were added in the last paragraph 6 (Conclusions and final remarks) of the paper:

Summarising, attention should be paid to the limitations that may affect the obtained results. The biggest limitation is that the analysis was performed in 2D instead of 3D. Conducting a two-dimensional analysis is a simplification both in the analytical and experimental setup sphere. In the analytical sphere, resignation from one dimension significantly simplifies the formulation, and in FEM analysis, it is possible to map the internal structure of the mixture in detail while maintaining a sufficient number of finite elements (415 x 415 = 172,215 elements). At the same level of detail, a 3D analysis would require a mesh of over 71 million elements.

- The chemical composition of the used materials are missing and must be added. It would be better to prepare a tabular form of the parameters that affects the behavior of the asphalt material.

Addressing this remark table 1 in paragraph 3.1 was split into two tables, and the chemical/petrographic origin of materials was added and presented in tabular form.

- The recommendation for the future work must be added

Addressing this remark, the following explanations were added in the last paragraph 6 (Conclusions and final remarks) of the paper:

The above comments regarding the limitations of the presented formulation are, at the same time, an indication of directions for further research. Of course, the correctness of this formulation should be verified by comparing the influence of 2D and 3D modelling on the obtained results (treating 3D models as a reference point). In addition, for 2D modelling, a procedure for preparing the surface for scanning should be proposed so that the air void ratio remains at the appropriate expected level. The natural direction of extending the methodology presented in this work is to repeat the analysis for other characteristic asphalt mixtures found in road construction, such as SMA, porous asphalt, etc., considering different asphalt binders (including highly modified binders).

Reviewer 3 Report

Attractive manuscript related to assessing the influence of mortar's Poisson ratio and viscous properties on stiffness and anisotropy of an asphalt concrete, including a very interesting application of the finite element method.

In general, the manuscript is well presented, but some details must be clarified or corrected/completed:

1.      Line 93: the expression “… 16 mm gradation …” is not the best. In this case, you used an “Asphalt Concrete with maximum aggregate size 16 mm …”;

2.      Again, line 93: the complete designation of the mixture (AC) must include the layer type and binder class. For example: “AC 16 base 50/70”;

3.      Figure 1: what does the "W" in “AC 16W” mean?

4.      Table 1: please confirm if your filler was a 0-0.063 mm. Did it not exceed 0.063 mm?

5.      Again, Table 1: the mentioned: “Density of mineral aggregate” (2.759) corresponded to what aggregate fraction?

6.      And again, Table 1: I prefer the designation “Voids filed by binder VFB” instead of “Voids filed by bitumen VFB”;

7.      Equations (1) and (2): please identify all the variables;

8.      Lines 190 and 241: please include the software version;

9.      Section “5. Results discussion”: you should include more discussion in light of results obtained by other researchers (if you find some similar and recent ones);

10.   Line 368: where you write “…to other types of asphalt mixtures because …” I would write “…to other types of asphalt mixtures and aggregates because …”;

11.   As these results should not be generalized to other mixtures (and aggregates), I would note this limitation immediately in the Abstract. It would be best if you also emphasised the fact that only a tiny portion of the mixture was analysed;

12.   In “References”, you can include the digital object identifier (DOI) for all references where available (as “encouraged” in the “Instructions for Authors”);

13.   A final remark: only 8 (in 25) references are less than five years old.

Author Response

Thank you very much for the quick review process and provided comments improving our paper. All of the comments have been reflected in the fixed manuscript.

- Attractive manuscript related to assessing the influence of mortar's Poisson ratio and viscous properties on stiffness and anisotropy of an asphalt concrete, including a very interesting application of the finite element method.

We thank the reviewer for their appreciation.

- Line 93: the expression “… 16 mm gradation …” is not the best. In this case, you used an “Asphalt Concrete with maximum aggregate size 16 mm …”;

Thank you for this remark. The text was improved as requested.

- Again, line 93: the complete designation of the mixture (AC) must include the layer type and binder class. For example: “AC 16 base 50/70”;

Thank you for this remark. The complete description of the mixture was added as requested: AC 16 binder layer PmB 25/55-60

- Figure 1: what does the "W" in “AC 16W” mean?

Thank you for this remark. The mixture description in figure 1 was modified and unified in the whole text.

- Table 1: please confirm if your filler was a 0-0.063 mm. Did it not exceed 0.063 mm?

Thank you for this question. Gradation of the filler was (89% passes #0.063, 100% passes #0.125). The clarification was added to the text in paragraph 3.1.

- Again, Table 1: the mentioned: “Density of mineral aggregate” (2.759) corresponded to what aggregate fraction?

Thank you for this question. Table 1 in paragraph 3.1 was split into tables 1 and 2. Densities presented in table 1 refer to the individual materials used to compose the mixture. In table 2, the density of mineral aggregate refers to the mixture of all aggregate in HMA.

- And again, Table 1: I prefer the designation “Voids filed by binder VFB” instead of “Voids filed by bitumen VFB”;

Thank you for this remark. The text was Improved as requested.

- Equations (1) and (2): please identify all the variables;

Thank you for this remark. The text was Improved as requested.

- Lines 190 and 241: please include the software version;

Thank you for this remark. The software versions were added to the text.

- Line 368: where you write “…to other types of asphalt mixtures because …” I would write “…to other types of asphalt mixtures and aggregates because …”;

Thank you for this remark. The text was Improved as requested.

- As these results should not be generalized to other mixtures (and aggregates), I would note this limitation immediately in the Abstract. It would be best if you also emphasised the fact that only a tiny portion of the mixture was analysed;

Thank you for this remark. At the end of the abstract, the following explanations were added as requested:

The obtained results are limited to particular asphalt concrete and shouldn’t be extrapolated to other asphalt mixture types without prior analysis.

- In “References”, you can include the digital object identifier (DOI) for all references where available (as “encouraged” in the “Instructions for Authors”);

Thank you for this remark. The bibliography was Improved on DOI as requested.

- Section “5. Results discussion”: you should include more discussion in light of results obtained by other researchers (if you find some similar and recent ones);

and

- A final remark: only 8 (in 25) references are less than five years old.

Thank you for this remark. We appreciate the reviewer's concern for the quality of the scientific discussion regarding recent papers. In Paragraph 5, some references and discussions were added as requested. Also, new items have been added to the reference, which better balances the previous references and the latest achievements in the subject of work.

Round 2

Reviewer 2 Report

thank you for implementing my previous comments and suggestions